# In Vitro Antimicrobial Activity of *Piper betle* Leaf Extract and Some Topical Agents against Methicillin-Resistant and Methicillin-Susceptible *Staphylococcus* Strains from Canine Pyoderma

**DOI:** 10.3390/ani12223203

**Published:** 2022-11-18

**Authors:** Patcharaporn Phensri, Kokaew Thummasema, Udomlak Sukatta, Serge Morand, Chantima Pruksakorn

**Affiliations:** 1Department of Microbiology and Immunology, Faculty of Veterinary Medicine, Kasetsart University, 50 Ngamwongwan Road, Bangkok 10900, Thailand; 2Kasetsart Agricultural and Agro-Industrial Product Improvement Institute, Kasetsart University, Bangkok 10900, Thailand; 3MIVEGEC, CNRS-IRD-Montpellier University, 911 Avenue Agropolis, 34394 Montpellier, France; 4Faculty of Veterinary Technology, Kasetsart University, Bangkok 10900, Thailand

**Keywords:** *Piper betle* leaf extract, azelaic acid, benzoyl peroxide, miconazole, chlorhexidine, dog, pyoderma, *Staphylococcus pseudintermedius*, *Staphylococcus schleiferi* subsp. *coagulans*

## Abstract

**Simple Summary:**

Pyoderma is one of the most common diseases in dogs. The primary pathogen isolated from canine pyoderma is *Staphylococcus pseudintermedius*, followed by *Staphylococcus schleiferi* subsp. *coagulans*. With the emergence of a multidrug-resistant strain, namely, methicillin-resistant staphylococci (MRS), topical antimicrobial therapy is encouraged. In this study, the in vitro antimicrobial activity of crude *Piper betle* leaf extract and some topical antimicrobials against clinical *S. pseudintermedius* and *S. schleiferi* subsp. *coagulans*, including their MRS strains, were evaluated. It was found that betel leaf extract demonstrated good antimicrobial activity with a higher proficiency than azelaic acid and benzoyl peroxide. Accordingly, betel leaf extract could provide a novel antimicrobial treatment, which may help reduce the need for systemic antibiotics, for canine pyoderma.

**Abstract:**

As multidrug-resistant methicillin-resistant staphylococci (MRS) is becoming more prevalent in canine pyoderma, the discovery of new therapeutic options is required. This study aimed to test the antimicrobial activity of crude *Piper betle* leaf extract and some topical antimicrobial agents against canine *Staphylococcus* clinical strains by determining the minimum inhibitory concentration (MIC) and the minimum bactericidal concentration (MBC). The results showed that the mean MICs of chlorhexidine, miconazole, crude *P. betle* leaf extract, azelaic acid, and benzoyl peroxide against *Staphylococcus* strains were 1.41, 1.62, 252.78, 963.49, and 1342.70 mg/L, respectively. Therefore, betel leaf extract demonstrated a superior efficacy to azelaic acid and benzoyl peroxide. Furthermore, the ratio of MBC/MIC of betel leaf extract was 1.75, indicating its bactericidal action. When applied to methicillin-resistant *S. pseudintermedius* (MRSP) and methicillin-susceptible *S. pseudintermedius* (MSSP), betel leaf extract was equally efficient towards both groups. *S. pseudintermedius* strains were more susceptible to betel leaf extract than *S. schleiferi* subsp. *coagulans*. In gas chromatography–mass spectrometry analysis, eugenol and hydroxychavicol appeared to be the major components of betel leaf extract. Given its efficacy, dogs with pyoderma could benefit from the use of betel leaf extract as a topical antimicrobial alternative.

## 1. Introduction

Pyoderma is a common dermatologic disease in dogs [1]. According to previous studies in the United Kingdom and Canada, pyoderma was found in 10.8% and 25.3% of canine dermatological disorders, respectively, which represents an estimated one fifth of the total dog caseloads [1,2]. In another UK survey, a rate of 1.3% pyoderma was reported among the total number of dogs presented to veterinary practitioners [3]. The pathogenesis of pyoderma is described as a bacterial skin infection, typically secondary to underlying primary disorders, including allergies, ectoparasites, and endocrinopathy, that weaken the skin defense mechanisms [4]. *Staphylococcus pseudintermedius* is the most common bacteria associated with canine pyoderma, along with, to a lesser extent, *Staphylococcus schleiferi* subsp. *coagulans* [5,6,7,8,9]. Antibacterial treatment usually relies on systemic antimicrobial therapy [3,10]. Among dogs with pyoderma, 91.9% were prescribed systemic antimicrobials, either alone or in combination with a topical product [3]. However, the growing number of methicillin-resistant staphylococci (MRS) cases has limited the use of many conventional antimicrobial agents [8,10,11,12]. MRS strains are resistant to all beta-lactam antibiotics, with the exception of the newest generation of cephalosporin beta-lactams [13,14]. An additional resistance to other drug classes (macrolides, aminoglycosides, fluoroquinolones, and sulfonamides) is frequently found among MRS isolates and is defined as multidrug resistance [8,15]. Some MRS strains are associated with zoonotic problems and pose a threat to public health [16]. The alarming antimicrobial resistance problem highlights the role of topical therapy in an antimicrobial stewardship strategy in treating canine pyoderma [5,17,18]. Furthermore, the global veterinary dermatology drug market is expected to expand by 9.5% and be worth nearly USD 10 billion by 2031 [19]. With the trend of pet adoption increasing, this is expected to be a key factor to bolster the market. New topological inventions could not only obtain more market value from other sectors but also drive market growth.

Topical therapy is recommended as the sole treatment or as an adjunctive to systemic antimicrobial agents for superficial pyoderma [5,18]. Chlorhexidine alone, chlorhexidine with miconazole, and benzoyl peroxide, in formulations such as shampoos, are preferred for generalized lesions [5]. With *Malassezia* infections, a combination of chlorhexidine and miconazole has been found to be clinically effective [20]. Topical compounds, including chlorhexidine, miconazole, benzoyl peroxide, fusidic acid, and mupirocin, have demonstrated anti-staphylococcal activity against canine isolates [21,22,23,24,25]. Chlorhexidine and miconazole’s MICs towards canine staphylococcal isolates have been found to be low, with their MIC_50_ values ranging from 0.5 to 8 mg/L, and differences in susceptibility related to geography could be observed [22,23,24]. A benzoyl peroxide-containing shampoo has been tested for its in vitro efficacy in canine *S. pseudintermedius*, with an MBC of 1:2 to 1:8 dilutions [25]. However, data on benzoyl peroxide’s MIC and MBC, which can be used to evaluate its efficacy in a more direct way, are lacking. For localized pyoderma lesions, a variety of hydroxyl acids represent a treatment option [5]. In human medicine, azelaic acid, a dicarboxylic acid, is regarded as a topical acne treatment [26]. Particularly, it is recommended as a second-line therapy [26]. The antimicrobial activity of azelaic acid has been documented in many bacteria, including pathogenic *Staphylococcus* species of human skin [27]. It has not been evaluated in terms of its anti-staphylococcal efficacy for veterinary purposes. 

In addition to the in vitro antimicrobial evidence, the clinical efficacy of chlorhexidine shampoo in canine superficial pyoderma has been evaluated to be the most effective, compared with benzoyl peroxide and ethyl lactate shampoos [17,28]. Due to the superior effectiveness of chlorhexidine, it is considered to be one of the most widely used topical agents [29]. Although the insensitivity of bacteria to chlorhexidine has been uncommon in veterinary dermatology, there is evidence suggesting that the use of chlorhexidine and the overall exposure of bacteria to chlorhexidine increases the risk of resistance not only to chlorhexidine, but also to some antimicrobials [30]. There is correspondingly a concern regarding the use of and resistance to fusidic acid and mupirocin in animals, which has led some European countries to reserve these drugs for human medicine [31]. Hence, there is an urgent need for additional treatment options. 

Natural remedies have become increasingly interesting in veterinary dermatology, with a focus on plants and their derivatives, which may provide alternative antimicrobial compounds [32,33]. Mangosteen extract, manuka honey, and many essential oils have been reported to have antimicrobial activity against *S. pseudintermedius* clinical isolates [34,35,36,37]. Among the essential oils is betel vine (*Piper betle* Linn.) oil [38]. As an alternative to its essential oil, *P. betle* can also be used as a crude extract, which has been reported to possess greater antibacterial activity [39]. Regarding its background, *P. betle* is a member of the *Piperaceae* family, which is widely cultivated in Thailand, Malaysia, Taiwan, India, and Sri Lanka [40]. In traditional medicine, the leaf is used to treat a wide range of health conditions and skin infections [39]. Recent research has shown that *P. betle* leaves possess antibacterial activity against a variety of Gram-negative and Gram-positive bacteria [41]. Based on studies of its essential oil, *P. betle* consists of a variety of phytochemically active compounds, classified as alkaloids/amides, propenylphenols, terpenes/sesquiterpenes, steroids, and prenylated hydroxybenzoic acids [42]. Several compounds from the propenylphenol group, such as hydroxychavicol and eugenol, have been detected in *P. betle* that likely contribute to its antibacterial properties [42,43]. Crude *P. betle* leaf extracts could be effective against *Staphylococcus* species of canine pathogens and provide a therapeutic option for dogs suffering from pyoderma. 

As benzoyl peroxide, miconazole, and chlorhexidine are standard topical treatments, they were included in this study. In addition, azelaic acid is not a dermatology primary option but another candidate for the treatment of canine pyoderma. This study aimed to assess the in vitro antimicrobial activity of crude *P. betle* leaf extract against canine *Staphylococcus* clinical isolates, including methicillin-resistant *S. pseudintermedius* (MRSP), methicillin-susceptible *S. pseudintermedius* (MSSP), and methicillin-resistant *S. schleiferi* subsp. *coagulans* (MRSS) and to compare it with that of azelaic acid, benzoyl peroxide, miconazole, and chlorhexidine. 

## 2. Materials and Methods

### 2.1. Bacterial Strains

Seventy-five *Staphylococcus* clinical isolates, including 31 methicillin-resistant *S. pseudintermedius* (MRSP), 21 methicillin-susceptible *S. pseudintermedius* (MSSP), 19 methicillin-resistant *S. schleiferi* subsp. *coagulans* (MRSS), 3 methicillin-susceptible *S. schleiferi* subsp. *coagulans* (MSSS), and one methicillin-susceptible *Staphylococcus simulans*, were obtained from a strain collection at the Department of Microbiology and Immunology, Faculty of Veterinary Medicine, Kasetsart University. These strains had been isolated from cases of canine pyoderma in the dermatology unit at Kasetsart University Veterinary Teaching Hospital, Bangkok, Thailand as part of the standard care of the patients and were not related to the present study. Brief details of the identification and characterization of these clinical strains are as follows. The species identification and methicillin resistance were confirmed by a polymerase chain reaction (PCR) based on *nuc* [44] and *mec*A [45] genes, respectively. Control strains for PCR included *S. aureus* ATCC 29213 (DMST 4745, Department of Medical Sciences, Thailand), the *mec*A-positive strain *S. pseudintermedius* Thai 36 [15], and *S. schleiferi* subsp. *coagulans* VETKU-SSC-5972. The *S. schleiferi* control was identified with PCR and sequencing. *S. simulans* were classified via MALDI-TOF-MS (Vitek^®^ MS, bioMérieux SA, Marcy-l’Étoile, France). The strains were stored in Luria–Bertani broth containing 20% glycerol (MilliporeSigma, Burlington, MA, USA) at −80 °C until use, and a culture at 37 °C for 18–24 h was performed on trypticase soy agar (Becton Dickinson, Franklin Lakes, NJ, USA). 

### 2.2. Preparation of Piper betle Leaf Extract and Analysis of Its Chemical Composition 

The betel leaf extract was prepared as a crude ethanolic extract. Briefly, betel leaves were purchased from a retail market in Nakhon Pathom Province, Thailand. After cleaning and air drying, the betel leaves were ground into a powder and then extracted via a maceration method using 95% ethanol as a solvent. The extract was condensed by using a rotary evaporator (Buchi Rotavapor R-124, Flawil, Switzerland). The gluey betel crude extract was kept in a desiccator at 4 °C.

The chemical constitution of the betel extract was characterized by gas chromatography and mass spectrometry (GC-MS model GC-2030 mass selective detector, Shimazu, Japan) at the Kasetsart Agricultural and Agro-Industrial Product Improvement Institute (Kasetsart University, Bangkok, Thailand). The GC part, with a column of DB-5MS 30 m × 0.25 mm ID × 0.25 mm, was injected with a sample volume of 1.0 µL (split ratio 1:20). Helium gas flowed into the column at a rate of 1.0 mL/min. The column temperature was programmed using an initial temperature of 60 °C for 3 min and then increasing at a speed of 10 °C/min to a temperature of 280 °C for 1 min. As for the MS operation, the ion source temperature was 250 °C in the electron impact ionization (EI) system. The extraction of the crude betel composition was obtained as a total ion chromatogram (TIC) in the scan mode using a range of mass of 35 to 500 atomic mass units. The identification of the query GC-MS mass spectrum was carried out by matching with the reference mass spectra of the Wiley Version 5 NIST 5 Library, together with a consideration of the Kovats retention index in the Adams table reference. The relative percentage was calculated by comparing its average peak area to the total area.

### 2.3. Determination of Minimum Inhibitory Concentration

The determination of the MIC was performed via a broth microdilution, according to the CLSI guidelines [46]. A stock solution was prepared at a 10× final concentration and adjusted for the drug’s potency or purity. *P. betle* leaf extract, fusidic acid (MilliporeSigma, Burlington, MA, USA), azelaic acid (MilliporeSigma, Burlington, MA, USA), benzoyl peroxide (MilliporeSigma, Burlington, MA, USA), miconazole nitrate (MilliporeSigma, Burlington, MA, USA), and chlorhexidine (MilliporeSigma, Burlington, MA, USA) were weighed and dissolved in different solvents at different concentrations (Table 1). The prepared solutions were clarified by filtration through a 0.2 µm syringe filter (Minisart, Darmstadt, Germany). The solvent controls were tested and it was shown that there was no interference with the bacteria growth. A working solution was serially diluted two-fold in cation-adjusted Mueller–Hinton broth (MHB, Becton Dickinson, Franklin Lakes, NJ, USA), and 100 µL of each dilution was added to a well of a 96-well polypropylene microplate (Corning, NY, USA). A bacterial suspension was prepared in a 0.85% sodium chloride (Vivantis, Shah Alam, Malaysia) solution, equivalent to the turbidity of the 0.5 McFarland standard, and further diluted to give a cell density approximately of 10^6^ colony forming units (CFU/mL). The aforementioned wells were then filled with a 10 µL volume of the prepared inoculum. The inoculum quantity control was performed by colony plating with the acceptable range of 2 × 10^5^ to 8 × 10^5^ CFU/mL [46]. Positive and negative growth control wells with MHB were included. Control wells without an inoculation to the serially diluted working reagent were also run in parallel. The plates were incubated at 37 °C for 18–24 h. The *S. pseudintermedius* Thai 36 was used as a quality control strain between independent assays. The MICs were determined as the lowest concentration of the bacterial test substance showing no visible growth. The reading of the microbial growth for MICs of betel extract, benzoyl peroxide, miconazole, and chlorhexidine was aided by the addition of 30 µL of resazurin (MilliporeSigma, Burlington, MA, USA) at a 0.02% concentration. The plates were incubated for a further 2 to 4 h [47]. Visible growth changed the blue dye (no growth) to a purple or pink color. The MIC values obtained by the colorimetric assay and visual observation were identical, with a difference of no more than one dilution. Due to azelaic acid’s reactivity with resazurin dye, the MIC was read by observing the broth’s clearness (no growth) and turbidity. Each isolate was assessed in triplicate and the discrepancy was accepted at only one dilution. The MIC test was performed concurrently for fusidic acid with the CLSI quality control strain, *S. aureus* ATCC 29213. According to the colorimetric and visual observations, the MIC values of fusidic acid against *S. aureus* ATCC 29213 were within the CLSI acceptable range [48].

### 2.4. Determination of Minimum Bactericidal Concentration

The determination of the MBC was carried out with the broth microdilution method [49]. Briefly, an aliquot of 10 µL from each well of the MIC-tested plates was spotted on the surface of TSA (Becton Dickinson, Franklin Lakes, NJ, USA) and incubated at 37 °C for 24 h. The culture plates were then counted for the bacterial colonies, and the lowest concentration showing a 99.9% killing of the final inoculum was recorded as the MBC.

### 2.5. Statistical Analysis

Analyses were performed using NCSS software version 2021 (NCSS, LLC, East Kaysville, UT, USA). The MIC and MBC data are presented as the means and standard deviations. The differences between the betel leaf extract, azelaic acid, benzoyl peroxide, miconazole, and chlorhexidine towards *Staphylococcus* isolates, and differences between *Staphylococcus* groups (MRSP, MSSP, and MRSS), were compared using non-parametric Kruskal–Wallis one-way ANOVA tests. The chosen level of significance for the differences between the testing reagents was *p* < 0.005 and that for the differences between *Staphylococcus* groups was *p* < 0.017, according to the Bonferroni correction.

## 3. Results

### 3.1. Phytochemicals of Crude Ethanolic Piper betle Extract

A phytochemical assay on *P. betle* ethanolic extract was performed using GC-MS analysis, as shown in Table 2. Eugenol was the predominant constituent, with a 44.17% peak area, followed by hydroxychavicol (26.34%). Chavicol, acetate (7.02%), γ-muurolene (5.27%), hexadecanoic acid, ethyl ester (3.60%), δ-cadinene (2.11%), 2,4 di-tert-butyl-phenol (1.84%), n-hexadecanoic acid (1.55%), and cis-calamenene (1.33%) were present in the lower percentages. Other components representing 6.78% of the peak area were 3-ethyl-3-methyl heptane, 5-methyl-5-propylnonane, trans-caryophyllene, α-selinene, α-calacorene, spathulenol, hexadecane, n-pentadecanal, tau-cadinol acetate, neophytadiene, and globulol. Since eugenol and hydroxychavicol are among the reported antibacterial compounds [43], the *P. betel* extract is considered to have an antibacterial potency in the majority proportion. The anti-staphylococcal activity of the betel leaf extract was then demonstrated.

### 3.2. Minimum Inhibitory Concentrations (MICs) and Minimum Bactericidal Concentrations (MBCs) of Piper betle Leaf Extract, Azelaic Acid, Benzoyl Peroxide, Miconazole, and Chlorhexidine against Staphylococcus Clinical Isolates of Canine Pyoderma

The antibacterial activity of betel leaf extract, azelaic acid, benzoyl peroxide, and miconazole against 75 canine *Staphylococcus* clinical isolates at the minimum inhibitory and bactericidal concentrations is shown in Figure 1. Chlorhexidine’s MIC towards 39 *Staphylococcus* isolates, including 16 MRSP, 11 MSSP, 10 MRSS, and 2 MSSS, is also included in the figure. As compared to azelaic acid (mean MIC 963.49 mg/L) and benzoyl peroxide (mean MIC 1342.70 mg/L), the betel leaf extract had a significantly lower MIC (mean 252.78 mg/L) (*p* < 0.0001). Miconazole and chlorhexidine demonstrated the lowest MICs (mean 1.62 and 1.41 mg/L, respectively) (*p* < 0.0001). The MICs of the betel leaf extract, azelaic acid, benzoyl peroxide, miconazole, and chlorhexidine ranged from 125 to 500, 500 to 1000, 250 to 2000, 0.39 to 3.12, and 0.5 to 2 mg/L, respectively. The MIC_50_ and MIC_90_ of the betel leaf extract were 250 mg/L, while azelaic acid and chlorhexidine displayed an MIC_50_ and MIC_90_ of 1000 and 1 mg/L, respectively. Miconazole and benzoyl peroxide exhibited only a two-fold difference in the values of the MIC_50_ (1.56 and 1000 mg/L, respectively) and MIC_90_ (3.12 and 2000 mg/L, respectively). The mean MBCs of the betel leaf extract, azelaic acid, benzoyl peroxide, and miconazole, as shown in Figure 1, were 443.06, 1348.68, 2070.32, and 6.55 mg/L, respectively, consistent with the MIC results. When the MBC/MIC ratio was considered, all the compounds except miconazole displayed an MBC/MIC ≤ 4, indicating bactericidal action. In addition, the MICs of the betel leaf extract, azelaic acid, benzoyl peroxide, and miconazole towards *S. aureus* ATCC 29213 were 500, 2000, 4000, and 1.56 mg/L, respectively (data not shown). 

### 3.3. Minimum Inhibitory Concentrations (MICs) of Piper betle Leaf Extract, Azelaic Acid, Benzoyl Peroxide, Miconazole, and Chlorhexidine against Methicillin-Resistant Staphylococcus pseudintermedius (MRSP), Methicillin-Susceptible Staphylococcus pseudintermedius (MSSP), and Methicillin-Resistant Staphylococcus schleiferi subsp. coagulans (MRSS)

The mean MICs and standard deviations of the betel leaf extract, azelaic acid, benzoyl peroxide, miconazole, and chlorhexidine were characterized according to the MRSP, MSSP, and MRSS groups (Figure 2). There was a statistically significant difference in the betel MICs, in which MRSS (mean 346.49 mg/L) showed a lower susceptibility than MRSP (mean 212.37 mg/L) and MSSP (mean 216.27 mg/L) (*p* < 0.0001). The mean MICs of azelaic acid towards MRSP, MSSP, and MRSS were 976.19, 920.63, and 982.46 mg/L, respectively. Benzoyl peroxide showed mean MICs against MRSP, MSSP, and MRSS of 1589.86, 1099.21, and 1175.44 mg/L, respectively. In addition, the MRSP group had a significantly higher benzoyl peroxide MIC than the MSSP and MRSS groups (*p* < 0.0001). Miconazole gave mean MICs for MRSP, MSSP, and MRSS of 1.55, 1.74, and 1.59 mg/L, respectively. The mean MICs of chlorhexidine towards MRSP, MSSP, and MRSS were 1.31, 1.23, and 1.75 mg/L, respectively. MRSP, MSSP, and MRSS were susceptible to azelaic acid, miconazole, and chlorhexidine, with no significant difference between these strains.

### 3.4. Minimum Bactericidal Concentrations (MBC) of Piper betle Leaf Extract, Azelaic Acid, Benzoyl Peroxide, and Miconazole against Methicillin-Resistant Staphylococcus pseudintermedius (MRSP), Methicillin-Susceptible Staphylococcus pseudintermedius (MSSP), and Methicillin-Resistant Staphylococcus schleiferi subsp. coagulans (MRSS)

The MBCs of the betel leaf extract, azelaic acid, benzoyl peroxide, and miconazole against MRSP, MSSP, and MRSS are shown in Table 3. The MBCs of the betel leaf extract for the MRSP and MSSP groups were 312.50 and 295.63 mg/L, respectively, while the MBC of 710.53 mg/L was obtained for the MRSS group. Azelaic acid showed MBCs in the MRSP, MSSP, and MRSS groups of 1250.38, 1185.19, and 1605.26 mg/L, respectively. The benzoyl peroxide MBCs in the MRSP, MSSP, and MRSS groups were 1898.62, 1908.73, and 2491.23 mg/L, respectively. Miconazole exhibited MBCs in the MRSP, MSSP, and MRSS groups of 4.80, 5.37, and 8.69 mg/L, respectively. There was no significant difference in the MBCs of all compounds in the MRSP and MSSP groups, while MRSS gave significantly higher MBCs for all compounds compared to the other groups (*p* < 0.0001 for betel leaf extract, azelaic acid, and miconazole and *p* < 0.017 for benzoyl peroxide). Regarding the MBC/MIC ratio for the MRSP, MSSP, and MRSS groups, the ratios of betel leaf extract, azelaic acid, and benzoyl peroxide were less than four. However, the miconazole ratio in the MRSS group was the highest, at 5.47. 

## 4. Discussion

The increased prevalence of multidrug-resistant (MDR) strains, the emergence of methicillin-resistant staphylococci (MRS) in canine pyoderma, and the limited choices of antimicrobial drugs have led to the necessity of the discovery of new therapeutic alternatives [8,10,11,12]. The use of a topical antimicrobial treatment is likewise recommended to improve the treatment efficiency and reduce the systemic antimicrobial use [5,18]. Meanwhile, certain topical antimicrobials pose concerns regarding their use and resistance in animals [30,31]. Plants and their antimicrobial derivatives could be studied to respond to the need for additional treatment options in veterinary dermatology.

In this study, based on the testing of the MIC and MBC, betel leaf extract appeared to be a promising candidate for an antimicrobial treatment in dogs with pyoderma. Overall, the relevant *Staphylococcus* isolates were susceptible to betel leaf extract, with an MIC of 252.78 mg/L, which is considered to have a significant antimicrobial activity cut-off of 1000 mg/L [50]. It was also found that betel leaf extract had a superior efficacy to azelaic acid and benzoyl peroxide. For MRSP and MSSP, betel leaf extract showed an inhibitory effect with no measurable difference in the susceptibility. While the inhibition of MRSS required a higher MIC than the other two groups, this concentration is still considered noteworthy. Based on an MBC/MIC ratio of less than four, the betel leaf extract is bactericidal towards MRSP, MSSP, and MRSS, consistent with previous studies of methicillin-resistant *Staphylococcus aureus* (MRSA) [51,52]. Although there has been no confirmation that cidal agents are better than static agents, bactericidal agents are preferred in immunocompromised patients.

Several studies have demonstrated the anti-staphylococcal activity of *P. betle* leaf extract against *S. aureus* [41,51,52]. The current study reported the remarkable antimicrobial activity of ethanolic betel leaf extract against clinical MRSP, MSSP, and MRSS isolates in the MIC range of 125 to 500 mg/L. In a previous study on MRSA, betel MICs of 0.31 to 2.5 mg/mL (310 to 2500 mg/L) were reported [52]. While there is an overlap in these MIC values with those obtained in the current study, the susceptibility to betel leaf extract among MRSP, MRSS, and MRSA requires future work. In the GC-MS-based analysis, eugenol and hydroxychavicol appeared to be the most abundant components in the betel leaf extract. These phenolic compounds have been identified as major antibacterial compounds of betel leaves [42,43]. In previous studies on *S. aureus*, the MICs of eugenol and hydroxychavicol pure compounds have been reported to be 100 to 400 mg/L, and 200 mg/L, respectively [53,54]. This corresponds with the current study that found a betel MIC of 252.78 mg/L and the eugenol and hydroxychavicol peak areas of 44.17% and 26.24%. Accordingly, the activity of the betel leaf extract against the canine *Staphylococcus* clinical isolates could be mainly attributed to the eugenol and hydroxychavicol constituents. The mechanism of action of eugenol against *S. aureus* by targeting the bacterial cell membrane has been observed [53]. In addition, a study of the microarray analysis has revealed that MRSA responded to eugenol by an upregulated gene expression in unique metabolic pathways for a bacterial survival [55]. However, differences in species susceptibility can be expected, which could lead to differences in the mechanism of inhibition. Research into the mechanism of action of eugenol, as well as hydroxychavicol on canine *Staphylococcus* species, could contribute to the discovery of new targets for drug development.

The relatively narrow MIC_50_ and MIC_90_ values suggested that a resistance to betel leaf extract, azelaic acid, benzoyl peroxide, miconazole, and chlorhexidine is unlikely. However, the statistically significant difference between MRSP, MSSP, and MRSS in response to the activity of benzoyl peroxide was interesting. Whether this represents the impact of the species or methicillin resistance on benzoyl peroxide’s susceptibility is unclear. The lowest MICs and highest efficacy among the tested reagents were found for miconazole and chlorhexidine. Miconazole’s MICs of 1.55 and 1.74 mg/L were demonstrated for MRSP and MSSP, respectively, which is consistent with previous reports [23,56]. Chlorhexidine’s MICs towards MRSP (1.31 mg/L) and MSSP (1.23 mg/L) in the current and previous reports are consistent [23,24]. Low MIC values of miconazole and chlorhexidine were also revealed against MRSS, whose MICs were not significantly different from those of MRSP and MSSP. Hence, the anti-staphylococcal activity of miconazole and chlorhexidine was further supported by this study. Regarding the bactericidal action, there was evidence for the betel leaf extract, azelaic acid, and benzoyl peroxide, but not for miconazole. Benzoyl peroxide and azelaic acid are known for their bactericidal activity, which corresponds with the present study [27]. However, the lack of bactericidal action of miconazole needs further elucidation.

In addition, the results of this study indicate that azelaic acid could be another acid candidate for the treatment of canine pyoderma. Although its efficacy was lower than that of chlorhexidine, miconazole, and betel leaf extract, the azelaic acid MICs of 500 to 1000 mg/L are well below the concentrations achievable with 20% azelaic acid applications. Furthermore, MRSP, MSSP, and MRSS were susceptible to azelaic acid with no significant difference.

## 5. Conclusions

To the best of the authors’ knowledge, this is the first study on the in vitro antibacterial activity of crude *P. betle* leaf extract against *S. pseudintermedius* and *S. schleiferi* subsp. *coagulans* of canine pyoderma isolates, including their methicillin-resistant strains. The antibacterial efficacy of betel leaf extract was superior to that of azelaic acid and benzoyl peroxide and was lower than that of chlorhexidine and miconazole. Information on MICs and MBCs regarding the use of betel leaf extract as a topical antibacterial agent was also provided. Further studies on the microbial spectrum, toxicity, product formulation, stability, and clinical efficacy should be undertaken prior to the application of betel leaf extract in canine pyoderma.

## Figures and Tables

**Figure 1 animals-12-03203-f001:**
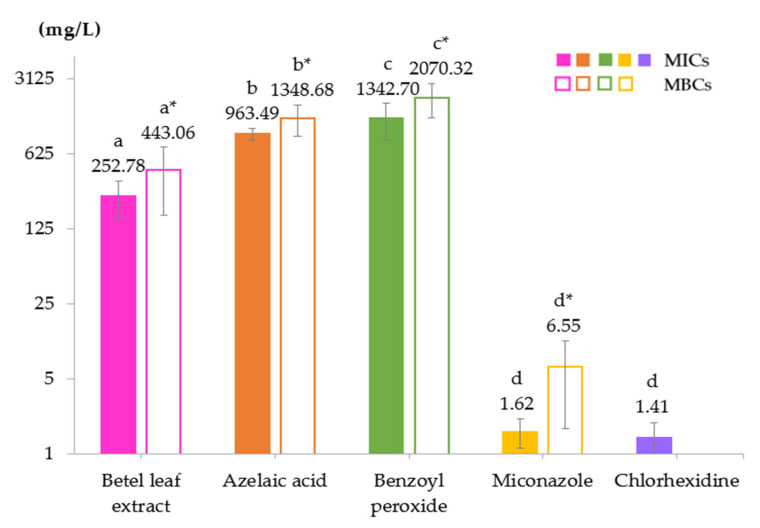
Minimum inhibitory concentrations (MICs) and minimum bactericidal concentrations (MBCs) of *Piper betle* leaf extract, azelaic acid, benzoyl peroxide, miconazole, and chlorhexidine against 75 *Staphylococcus* clinical isolates of canine pyoderma. Only chlorhexidine’s MIC was determined in 39 *Staphylococcus* isolates, and its MBC was not determined. Data were expressed as means and standard deviations. ^a, b, c, d^ Above mean MIC bars and ^a*, b*, c*, d*^ above mean MBC bars carrying different superscripts are significantly different by Kruskal–Wallis one-way ANOVA (*p* < 0.0001).

**Figure 2 animals-12-03203-f002:**
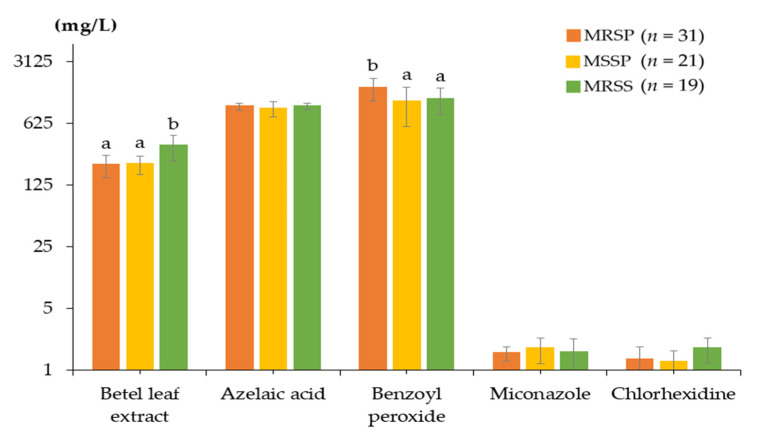
Minimum inhibitory concentrations (MICs) of *Piper betle* leaf extract, azelaic acid, benzoyl peroxide, miconazole, and chlorhexidine against methicillin-resistant *Staphylococcus pseudintermedius* (MRSP), methicillin-susceptible *Staphylococcus pseudintermedius* (MSSP), and methicillin-resistant *Staphylococcus schleiferi* subsp. *coagulans* (MRSS). Only chlorhexidine’s MIC was determined in 16 MRSP, 11 MSSP, and 10 MRSS. Data were expressed as means and standard deviations. ^a, b^ above mean MIC bars carrying different superscripts is significantly different by Kruskal–Wallis one-way ANOVA (*p* < 0.0001).

**Table 1 animals-12-03203-t001:** Stock solutions prepared.

	Stock Solution Concentration (mg/L)	Solvents
*P. betle* leaf extract	320,000	100% DMSO
Fusidic acid	20,480	Water
Azelaic acid	640,000	5% Tween 80 and 5% ethanol
Benzoyl peroxide	320,000	100% DMSO
Miconazole	4000	100% DMSO
Chlorhexidine	1280	Water

**Table 2 animals-12-03203-t002:** Phytochemical composition of *Piper betle* ethanolic leaf extract.

Compounds	Retention Time	Kovats Retention Index	Peak Area (%)
Chavicol, acetate	9.875	1195	7.02
Eugenol	11.675	1392	44.17
Hydroxychavicol	12.970	1424	26.34
γ-Muurolene	13.220	1478	5.27
2,4 di-tert-*butyl*-*phenol*	13.475	-	1.84
δ-Cadinene	13.740	1523	2.11
cis-calamenene	13.840	1528	1.33
n-hexadecanoic acid	18.605	1968	1.55
Hexadecanoic acid, ethyl ester	18.965	1978	3.60
Others	-	-	6.78

**Table 3 animals-12-03203-t003:** Minimum bactericidal concentrations (MBCs) of *Piper betle* leaf extract, azelaic acid, benzoyl peroxide, and miconazole against methicillin-resistant *Staphylococcus pseudintermedius* (MRSP), methicillin-susceptible *Staphylococcus pseudintermedius* (MSSP), and methicillin-resistant *Staphylococcus schleiferi* subsp. *coagulans* (MRSS).

Reagents	MRSP (*n* = 31)		MSSP (*n* = 21)		MRSS (*n* = 19)	
	MBC (mg/L) Mean ± SD	MBC/MIC Ratio	MBC (mg/L) Mean ± SD	MBC/MIC Ratio	MBC (mg/L) Mean ± SD	MBC/MIC Ratio
Betel leaf extract	312.50 ± 119.68 ^a^	1.47	295.63 ± 100.26 ^a^	1.37	710.53 ± 295.76 ^b^	2.05
Azelaic acid	1250.38 ± 382.93 ^a^	1.28	1185.19 ± 357.17 ^a^	1.29	1605.26 ± 462.18 ^b^	1.63
Benzoyl peroxide	1898.62 ± 485.89 ^a^	1.19	1908.73 ± 881.09 ^a^	1.74	2491.23 ± 772.75 ^b^	2.12
Miconazole	4.80 ± 3.23 ^a^	3.10	5.37 ± 2.99 ^a^	3.09	8.69 ± 4.99 ^b^	5.47

^a, b^ Mean values within the same row carrying different superscripts are significantly different by Kruskal–Wallis one-way ANOVA (*p* < 0.0001 for betel leaf extract, azelaic acid, and miconazole; *p* < 0.017 for benzoyl peroxide).

## Data Availability

The data presented in this study are available on request from the corresponding author.

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
