# Peer review of "In Vitro Antimicrobial Activity of Piper betle Leaf Extract and Some Topical Agents against Methicillin-Resistant and Methicillin-Susceptible Staphylococcus Strains from Canine Pyoderma"

_animals, 2022, doi:10.3390/ani12223203_

Round 1
Reviewer 1 Report
Dear Authors, your work is simple and clear. I am only skeptic about the limited number of tests employed, but the work is correctly done, and the results clearly presented. Otherwise, the text is plenty of typos and unproper use of English, that I would kindly as for a throughout revision.
Thea authors should emphasize also how big is the problem of canine pyoderma. Also investigating and reporting some data of the market size and values would beneficiate the papers to contextualize the application proposed.
Other minor revs
Line 15: often found the infection is Staphylococcus pseudintermedius and Staphylococcus schleiferi subsp. coagulans…Please revise this sentence
Line 22: which will reduce the need for systemic antibiotics…please revise the verbal tense, and use “could” or something like.
Abstract contains too much results
Line 24-25: the discovery of new therapeutic options is imperative…imperative seems excessive
Keywords: to gain higher indexing, please avoid to use those already present in the title
Introduction: this section is fine and easy to read, maybe a bit too narrative
Line 58: In addition to the in vitro evidence…are you referring to microbiological test only, so please specify
Line 76: Piperaceae…please report in italic
Materials and methods please generally correct typos like 280°C
Author Response
Response to reviewer 1 comments
Point 1: Dear Authors, your work is simple and clear. I am only skeptic about the limited number of tests employed, but the work is correctly done, and the results clearly presented. Otherwise, the text is plenty of typos and unproper use of English, that I would kindly as for a throughout revision.
Response 1: Thank you for your suggestion. The manuscript was completed for English editing by the MDPI service. Typos were carefully checked and corrected. Regarding the limited number of tests mentioned, the results of chlorhexidine’s MIC were also included in this revised manuscript. Other parts of the manuscript were revised, according to the chlorhexidine’s including as well.
Point 2: The authors should emphasize also how big is the problem of canine pyoderma.
Response 2: We have revised the manuscript to explain the question (Lines 44-48).
“According to previous studies in the United Kingdom and Canada, pyoderma was found in 10.8% and 25.3% of canine dermatological disorders, respectively, which represents an estimated one fifth of the total dog caseloads [1,2]. In another UK survey, a rate of 1.3% pyoderma was reported among the total number of dogs presented to veterinary practitioners [3].”
Point 3: Also investigating and reporting some data of the market size and values would beneficiate the papers to contextualize the application proposed.
Response 3: We have revised the manuscript, according to the suggestion (Lines 64-68).
“Furthermore, the global veterinary dermatology drug market is expected to expand by 9.5% and be worth nearly USD 10 billion by 2031 [19]. With the trend of pet adoption increasing, this is expected to be a key factor to bolster the market. New topological inventions could not only obtain more market value from other sectors but also drive market growth.”
Other minor revs
Point 4: Line 15: often found the infection is Staphylococcus pseudintermedius and Staphylococcus schleiferi subsp. coagulans…Please revise this sentence
Response 4: The sentence was revised as in Lines 15-17.
“Pyoderma is one of the most common diseases in dogs. The primary pathogen isolated from canine pyoderma is Staphylococcus pseudintermedius, followed by Staphylococcus schleiferi subsp. coagulans.”
Point 5: Line 22: which will reduce the need for systemic antibiotics…please revise the verbal tense, and use “could” or something like.
Response 5: The verbal tense was changed to “may help” (Line 23).
“Accordingly, betel leaf extract could provide a novel antimicrobial treatment, which may help reduce the need for systemic antibiotics, for canine pyoderma.”
Point 6: Abstract contains too much results
Response 6: The abstract was revised with the main results (Lines 25-39).
Point 7: Line 24-25: the discovery of new therapeutic options is imperative…imperative seems excessive
Response 7: The word “imperative” was changed to “required” (Line 26).
“As multidrug-resistant methicillin-resistant staphylococci (MRS) is becoming more prevalent in canine pyoderma, the discovery of new therapeutic options is required.”
Point 8: Keywords: to gain higher indexing, please avoid to use those already present in the title
Response 8: Keywords that do not present in the title were added.
The included keywords are “Piper betel leaf extract; azelaic acid; benzoyl peroxide; miconazole; chlorhexidine; dog; pyoderma; Staphylococcus pseudintermedius; Staphylococcus schleiferi subsp. coagulans” (Lines 40-41).
Point 9: Introduction: this section is fine and easy to read, maybe a bit too narrative
Response 9: Thank you for your suggestion. The introduction was revised by adding details, scientific data, and a conclusion throughout the part (Lines 44-48, 50, 54-55, 57-61, 70-73, and 75-87).
Point 10: Line 58: In addition to the in vitro evidence…are you referring to microbiological test only, so please specify
Response 10: Yes, we refer to a microbiological test and the sentence was revised (Lines 88-90).
“In addition to the in vitro antimicrobial evidence, the clinical efficacy of chlorhexidine shampoo in canine superficial pyoderma has been evaluated to be the most effective, compared with benzoyl peroxide and ethyl lactate shampoos [17,28]”.
Response 11: Line 76: Piperaceae…please report in italic
Response 11: It was revised in italic (Line 106).
“Regarding its background, P. betle is a member of the Piperaceae family, which is widely cultivated in Thailand, Malaysia, Taiwan, India, and Sri Lanka [40].”
Response 12: Materials and methods please generally correct typos like 280°C
Response 12: The kinds of typos (Lines 159, 186) were corrected throughout the Materials and methods.

Reviewer 2 Report
This manuscript by Phensri et al describes the activity of Piper betle leaf extract as a potential topological agent for use against various staphylococci that can cause canine pyoderma, which is a significant veterinary problem.
The data consist of two sets. First we are presented with analysis of the composition of Piper betle leaf extract, but there is no further use of this data - the authors do not attempt to determine the relative anti staphylococcal activities of these compounds and I feel that this is a missed opportunity.
Second comes analysis of the activity of Piper betle leaf extract against a panel of staphylococcal species, where it is compared to a selection of other antimicrobials. I feel that the authors need a better explanation of why these other compounds were chosen for comparison. Moreover, the data in tables 3 and 4 seem to be from the same assays, presented in two different ways. I feel these data would be better presented graphically as they would give the reader a better impression of the variability of the antimicrobial activity observed.
It would have been good to see some analysis of the activities of compounds found in the Piper betle abstract.
In the Discussion, the authors remind the reader that raw MIC data are not a good indicator of clinical efficacy and suitability, other things like marrow spectrum activity against other microorganisms is desirable, and things like compound solubility, stability and toxicity are very important too.
Author Response
Response to reviewer 2 comments
Comments and Suggestions for Authors
This manuscript by Phensri et al describes the activity of Piper betle leaf extract as a potential topological agent for use against various staphylococci that can cause canine pyoderma, which is a significant veterinary problem.
Point 1: The data consist of two sets. First we are presented with analysis of the composition of Piper betle leaf extract, but there is no further use of this data - the authors do not attempt to determine the relative anti staphylococcal activities of these compounds and I feel that this is a missed opportunity.
Response 1: Thank you for your suggestions. We have revised the manuscript in the Results (Lines 229-232) and Discussion (Lines 349-361) sections with the results of betel leaf extract composition relative to anti-staphylococcal activity.
In the Result, “Since eugenol and hydroxychavicol are among the reported antibacterial compounds [43], the P. betel extract is considered to have antibacterial potency in the majority proportion. The anti-staphylococcal activity of betel leaf extract was then demonstrated.”
In the Discussion, “In previous studies on S. aureus, the MICs of eugenol and hydroxychavicol pure compounds have been reported to be 100 to 400 mg/L, and 200 mg/L, respectively [53,54]. This corresponds with the current study that found a betel MIC of 252.78 mg/L and the eugenol and hydroxychavicol peak areas of 44.17% and 26.24%. Accordingly, the activity of betel leaf extract against the canine Staphylococcus clinical isolates could be mainly attributed to the eugenol and hydroxychavicol constituents. The mechanism of action of eugenol against S. aureus by targeting the bacterial cell membrane has been observed [53]. In addition, a study of microarray analysis has revealed that MRSA responded to eugenol by upregulated gene expression in unique metabolic pathways for bacterial survival [55]. However, differences in species susceptibility can be expected, which could lead to differences in the mechanism of inhibition. Research into the mechanism of action of eugenol, as well as hydroxychavicol on canine Staphylococcus species, could contribute to the discovery of new targets for drug development.”
Point 2: Second comes analysis of the activity of Piper betle leaf extract against a panel of staphylococcal species, where it is compared to a selection of other antimicrobials. I feel that the authors need a better explanation of why these other compounds were chosen for comparison.
Response 2: The Introduction section was revised to clarify the question in Lines 117-119 and with relevant background in Lines 70-87.
Lines 117-119: “As benzoyl peroxide, miconazole, and chlorhexidine are standard topical treatments, they were included in this study. In addition, as azelaic acid is not a dermatology primary option, but another candidate for the treatment of canine pyoderma.”
According to comments of Reviews 1 and 3, chlorhexidine was also included to compare with betel leaf extract, azelaic acid, benzoyl peroxide, and miconazole in this revised manuscript. Other parts of the manuscript were revised, according to this change.
Point 3: Moreover, the data in tables 3 and 4 seem to be from the same assays, presented in two different ways. I feel these data would be better presented graphically as they would give the reader a better impression of the variability of the antimicrobial activity observed.
Response 3: Thank you for your suggestion. The data in Tables 3 and 4 were changed to bar graphics of Figures 1 (Lines 259-264) and 2 (Lines 284-290) to show the results in a more visual way.
Point 4: It would have been good to see some analysis of the activities of compounds found in the Piper betle abstract.
Response 4: The abstract was revised with the results of GC-MS-based analysis of P. betle extract (Lines 37-38).
“In gas chromatography–mass spectrometry analysis, eugenol and hydroxychavicol appeared to be the major components of betel leaf extract.”
Point 5: In the Discussion, the authors remind the reader that raw MIC data are not a good indicator of clinical efficacy and suitability, other things like marrow spectrum activity against other microorganisms is desirable, and things like compound solubility, stability and toxicity are very important too.
Response 5: According to the comment, the Conclusions section was revised as in Lines 392-394.
“Further studies on the microbial spectrum, toxicity, product formulation, stability, and clinical efficacy should be undertaken prior to the application of betel leaf extract in canine pyoderma.”

Reviewer 3 Report
The paper by Pruksakorn et al concerns the antibacterial activity of betel leaf extract, azelaic acid, benzoyl peroxide, and miconazole in the treatment of Canine Pyoderma. As this condition can be caused by several different bacteria and has an increasing degree of anti-biotic resistance new treatments are of interest.
As noted, at length, by the authors the gold standard for such treatment is chlorhexidine. This leads to a major problem - why was chlorhexidine not studied? A number of unrelated antiseptics are compared to betel leaf extracts - why?
I found the paper confused and confusing, and it must be restructured with reference to chlorhexidine included before re-consideration.
Author Response
Response to reviewer 3 comments
Comments and Suggestions for Authors
The paper by Pruksakorn et al concerns the antibacterial activity of betel leaf extract, azelaic acid, benzoyl peroxide, and miconazole in the treatment of Canine Pyoderma. As this condition can be caused by several different bacteria and has an increasing degree of anti-biotic resistance new treatments are of interest.
Point 1: As noted, at length, by the authors the gold standard for such treatment is chlorhexidine. This leads to a major problem - why was chlorhexidine not studied? A number of unrelated antiseptics are compared to betel leaf extracts - why?
Response 1: Thank you for your questions. Chlorhexidine’s MIC against canine Staphylococcus was included in the revised version to complete the comparison. The data of chlorhexidine are still unpublished and concern 39 Staphylococcus isolates with species distribution and proportion (16 MRSP, 11 MSSP, and 10 MRSS) comparable to the 75 Staphylococcus isolates. In addition, chlorhexidine’s MIC was determined with the same standard method, as detailed in the Materials and Methods section. However, chlorhexidine’s MBC is not determined.
The Introduction section was revised to explain why these compounds (azelaic acid, benzoyl peroxide, miconazole, and chlorhexidine) were chosen for comparison in Lines 117-119 and with relevant background in Lines 70-87.
Lines 117-119: “As benzoyl peroxide, miconazole, and chlorhexidine are standard topical treatments, they were included in this study. In addition, as azelaic acid is not a dermatology primary option, but another candidate for the treatment of canine pyoderma.”
Other parts of the manuscript were revised, according to this change.
Point 2: I found the paper confused and confusing, and it must be restructured with reference to chlorhexidine included before re-consideration.
Response 2: We follow this comment and other comments of Reviewer 1 and Reviewer 2. We have greatly improved the manuscript to make it easier to read and follow.
We have added references to chlorhexidine: Reference 5, 20, 22-25.
